# Digital Transformation in Epilepsy Diagnosis Using Raw Images and Transfer Learning in Electroencephalograms

**Marlen Sofía Muñoz** [1], **Camilo Ernesto Sarmiento Torres** [1], **Ricardo Salazar-Cabrera** [2], **Diego M. López** [2,*] and **Rubiel Vargas-Cañas** [1]

1. Dynamic Systems, Instrumentation, and Control Research Group, Physics Department, Universidad del Cauca, Popayán 190001, Colombia
2. Telematics Engineering Research Group (GIT), Telematics Department, Universidad del Cauca, Popayán 190001, Colombia
* Correspondence: dmlopez@unicauca.edu.co

**Abstract:** Epilepsy diagnosis is a medical care process that requires considerable transformation, mainly in developed countries, to provide efficient and effective care services taking into consideration the low number of available neurologists, especially in rural areas. EEG remains the most common test used to diagnose epilepsy. In recent years, there has been an increase in deep learning techniques to analyze electroencephalograms (EEG) to detect epileptiform events. These types of techniques support the epilepsy diagnostic processes performed by neurologists. There have been several approaches such as biomedical signal processing, analysis of characteristics extracted from the signals, and image analysis to detect epileptiform events. Most of the works reported in the literature, which use images, transformed the signals into a two-dimensional space interpreted as an image. However, only a few of them use the raw EEG image. This paper presents a computational model for detecting epileptiform events from raw EEG images, using convolutional neural networks and a transfer learning approach. To perform this work, 100 pediatric EEGs were collected, noting six characteristics of epileptiform events in each exam: spikes, poly-spikes, spike-and-wave, sharp waves, periodic, and a combination of them. Then, pre-trained convolutional neural networks were used, which, through transfer learning techniques, were retrained to classify possible events. The model's performance was evaluated in terms of precision, accuracy, and Mathews' correlation coefficient. The model offered a performance above 95% accuracy for binary classification and above 87% for multi-class classification. These results demonstrated that identifying epileptiform events from raw EEG images combined with deep learning techniques such as transfer learning is feasible. Significance: The proposed method for the evaluation of EEG tests, as a support tool for the diagnosis of epilepsy, can help to reduce the time of reading EEGs, which is very important, especially in developing countries with a limitation of a specialist in neurology.

**Keywords:** AlexNet; digital transformation; EEG raw images; epilepsy diagnosis; pediatrics epilepsy; seizure detection; transfer learning

## 1. Introduction

Epilepsy is a functional cerebral disease caused by sudden abnormal brain neuron discharge. It is one of the most common brain diseases [1]. Multichannel electroencephalogram (EEG) has been widely applied for epilepsy analysis and diagnosis as it contains rich information on the abnormal discharge of brain cells during seizure onsets [2]. The EEG allows observing brain activity by using electrodes in the scalp's area and usually adopting the international system 10–20 [3]. The EEG is analyzed by a neurologist, who looks for electrographic characteristic events that represent epilepsy, also called epileptiform events, such as spikes, sharp waves, slow waves, etc. In addition, the specialist's task is also to identify non-epileptiform events [4], such as eye blinks, normal background activity, and

noise from different sources. However, analyzing EEG is a time-consuming and rigorous task since abnormalities such as spikes are only 20–70 milliseconds in length and cannot become unnoticed [5]. Therefore, performing an automatic EEG analysis is essential to approach the enormous challenge of supporting, facilitating, and expediting the diagnosis of epilepsy, especially in developing countries with a limitation of a specialist in neurology [6].

A developing country is defined as a country that has an annual gross national product per capita (GNP) of less than 9361 American dollars, according to the World Bank. Most low and middle-income countries (LMIC) fall into this category [7]. The number of neurologists available in LMIC is low causing consequences such as the low coverage of health services for epilepsy [8]. In Colombia, only 208 neurologists were available in 2017 (around one neurologist for every 240,000 inhabitants), while in developed countries the number of neurologists available is approximately ten times more [9]. It is also worrying that the expected number of neurologists for the year 2030, according to the governing entity of health in Colombia, would only reach 629, which is not enough to have a significant improvement [9]. The lack of available specialists in LMIC affects the epilepsy process and diagnosis due to the required time to analyze the diagnostic test.

Considering the low availability of neurologists in LMIC countries, digital transformation in epilepsy diagnosis is essential to support the process performed by neurologists and try to reduce the time needed to review an EEG exam, which is used as the principal tool for an epilepsy diagnosis. The time needed to review an EEG exam currently in developing countries is too long, approximately 30 to 60 min, which considerably limits the number of diagnoses per day that a neurologist can perform [5]. The use of information and communication technologies (ICT) is essential to support these diagnoses, reducing the time the neurologist spends reviewing an exam, and reducing the complexity of the process. By reducing the review time of an EEG exam, the sustainability of the service is improved in the social and economic aspects, allowing an increase in the number of patients diagnosed per day. The technologies used in this study have focused on trying to reduce the number of signals to be reviewed by the neurologist, because a page of an EEG exam (which lasts approximately 30 min), which normally has only 10 s of the exam and a large number of channels (one signal per channel) to review (an exam is approximately 180 pages) [5]. By identifying the possible events of interest in an EEG to a certain number (which is normally low), through ICT, the time needed to review the entire exam decreases considerably.

Machine learning (ML) algorithms are becoming a relevant tool to support the automatic detection of relevant information in EEG records. The objectives are diverse: recognition of emotions, evaluation of the sleep quality, and detection of epileptiform events, among others [2,10–16]. Approaches to detect epileptiform events using ML include biomedical signal processing, analysis of characteristics extracted from the signals, and analysis of images in a lesser proportion [17]. The works of Molina et al. [5] and Muñoz et al. [18] provide an example of the use of ML for the detection of epileptiform events. Molina et al. developed an intelligent component to automatically detect abnormal segments of EEG tests using conventional machine learning algorithms over a dataset generated from EEG signals [5]. Muñoz et al. present a machine learning-based methodology using a visual bag of words taken from raw EEG images as input to identify images with abnormal signals [18]. Although ML provides exciting results, it has many drawbacks due to the many channels used, the low amplitude per channel, and the non-stationarity of each channel in the signal [19].

The deep learning technique called transfer learning has been recently used to detect epileptiform events in EEG [20–24]. Transfer learning is the process of taking a pre-trained deep learning network and fine-tuning it to learn a new task [24]. Transfer learning algorithms use datasets, features, or model parameters from the source domain to train the model in the target domain to reduce the scale of training data in the target domain, reducing the sampling and training cost [25]. Qu et al. propose an epileptogenic region detection based on a deep Convolutional Neural Network (CNN) with transfer learning, which aided the automatic detection and classifications of focal signals from non-focal

signals [23]. Cao et al. present a comprehensive study on epileptic state classification based on deep transfer learning (TL) [20]. Nogay et al. propose an end-to-end machine learning model to detect epileptic seizures using the pre-trained deep two-dimensional CNN and the concept of transfer learning [22]. Gómez et al. propose an automatic method to detect epileptic seizures using an imaged-EEG representation of brain signals, analyzing EEG signals from two different datasets: the CHB-MIT Scalp EEG database and scalp and intracranial recordings of the EPILEPSIAE project [21]. Finally, Raghu et al. present a classifier of seven seizures with non-seizure EEG, which is developed by applying CNN and transfer learning using the Temple University Hospital EEG corpus (open-source database) [24].

Most works reported in the literature for detecting epileptiform events that use images, CNN, and transfer learning; performed a transformation of the signals to a two-dimensional space that can be interpreted as an image. However, to the best of our knowledge, few works use the raw EEG image (only three works in the researched literature). This research presents a computational model for detecting epileptiform events from raw EEG images and convolutional neural networks (pre-trained using transfer learning). This proposal seeks to improve the efficiency in the diagnosing epilepsy through a technology that allows the neurologist to reduce the time to review an EEG exam.

For the development of the proposal, first, 100 pediatric EEGs were collected to perform this work, noting six types of epileptiform events in each exam: sharp waves, spikes, poly-spikes, spike-and-wave, periodic, and combinations of the above. Next, 100 other pediatric EEGs were collected [5]. Then, pre-trained convolutional neural networks were used, which, through transfer learning techniques, were retrained to classify possible events. Finally, the model's performance was evaluated in precision, accuracy, sensitivity/recall, specificity, F1-score, and Mathews' correlation coefficient (MCC).

The novelty of our work is represented in the following aspects: (a) The use of raw images for abnormalities detection in EEG exams, which has been done in a few works worldwide. (b) Some of the works with similar approaches (using raw images/or transfer learning, and/or CNN) perform one of the two approaches we used in our research, binary classification or multi-class classification. We did not find works with both types of approaches. (c) The results presented by the works with similar approaches do not have sufficient performance indices. They only present some of them (accuracy and precision). In our work, we present the following performance indices: accuracy, sensitivity, specificity, precision, F1- score, and Matthews Correlation Coefficient (CCM). (d) The use of a pediatric EEG exams dataset in our research can be considered a novel result since few datasets on pediatric EEG exist, and pediatric EEG presents more significant variability than adult EEGs. Therefore, they have a greater level of difficulty in interpreting their signals.

The rest of this paper is divided into four more sections: Section 2 presents the materials and methods used in this research; Section 3 shows the results obtained in each stage of the methodology; Section 4 discusses the results, and, finally, Section 5 concludes this paper and presents some future work.

## 2. Materials and Methods

Transfer learning is part of the deep learning approach and is characterized by a recognition system that applies previously learned knowledge and skills to novel tasks [26]. It is commonly used to solve problems in implementing deep learning algorithms with small datasets, where a large volume of data is required for the training stage [27,28]. It is applied in two settings (Figure 1), taking a pre-trained model and adapting it to a new data set.

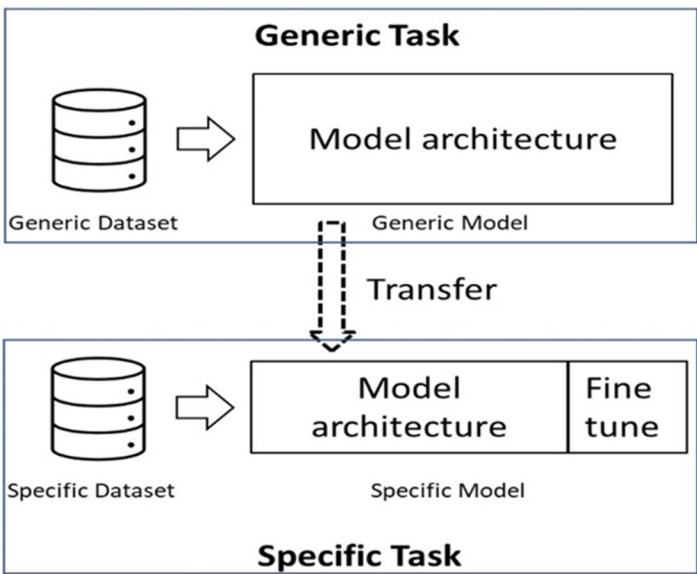

**Figure 1.** Block diagram of the transfer learning approach.

The principal material used to collect data within this research was the BWII EEG device (by Neurovirtual) provided with the BWII Analysis software (https://neurovirtual.com/). The device was set up for the 10–20 standard at a sampling frequency of 200 Hz, with a 50/60 Hz filter and digital filter provided by the manufacturer's software. Furthermore, the Ethics Committee of the University of Cauca, Colombia, consented to each EEG record in compliance with the Declaration of Helsinki and bioethical standards.

Concerning methods, the work performed in this research was divided into two stages. First, a binary classification (epileptic seizure or normal) of the analyzed images was conducted in the first stage. Then, a multi-class classification was performed in the second stage, identifying up to seven different classes: six possible types of an epileptic seizure and the normal signal. The methods used in each stage are described below.

### 2.1. First Stage: Binary Classification

At this stage, the methodology consisted of five steps (Figure 2): data collection, data preparation, model selection, model fine-tuning, and model evaluation.

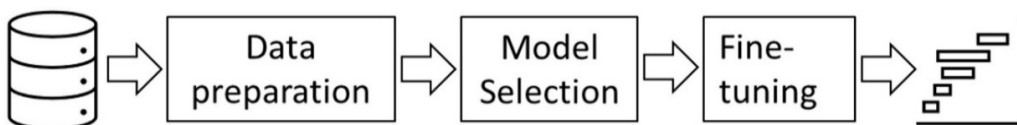

**Figure 2.** Five steps of the methodology proposed for stage 1.

The first step consisted of collecting and annotating pediatric EEG exams. Secondly, each EEG was segmented into its constituting pages, stored as individual images. Then, a generic deep learning model (pre-trained with a generic database) was selected. Next, a new model was built by transferring the generic knowledge to a specific data set. Finally, the performance of the model was evaluated with images of unseen patients.

#### 2.1.1. Data Collection: Acquisition of Encephalograms

To perform this work and others related [5,18,29–31], 100 pediatric EEGs were collected from the same number of children aged between 22 days and 17 years old, suspected of suffering epilepsy. The exams were performed with the patients asleep, on the recommendation of the neurologist. To achieve this, parents were asked to take the child to the test in sleep deprivation, sleeping between five and six hours the night before the appointment by delaying the child's bedtime and moving the wake time up.

Later, the exams were interpreted by pediatric neurologists noting six types of epileptiform events in each exam: sharp waves, spikes, poly-spikes, spike-and-wave, periodic, and combinations. This collected dataset was one of the objectives of the NeuroMoTIC project. This system helps with the diagnosis of epilepsy, data collection, management, and classification of clinical information and EEG signals [5].

In total, 100 EEGs were collected with a duration of 30 min per examination. At the same time, the interpretation and annotation of events captured were performed with the advice and supervision of a pediatric neurologist, who identified the main characteristics of the patients in the entire data set. In addition, these data were divided demographically, and the diagnosis between normal and abnormal is presented in Table 1.

**Table 1.** Patient demographics in the dataset.

| Number of Patients | Min. Age | Max. Age | Mean Age | Gender | | Conclusion |
|---|---|---|---|---|---|---|
| | | | | M | F | |
| 34 | 24 days | 14 years | 5.86 years | | | Abnormal |
| | | | | 21 | 13 | |
| 66 | 22 days | 17 years | 6.6 years | 39 | 27 | Normal |

### 2.1.2. Data Preparation

Events were divided into normal and abnormal (binary classification). Then, all EEGs were converted, page by page, to the European Data Format (EDF) format and stored, as images, using the EDFBrowser (https://www.teuniz.net/edfbrowser/). This tool is a free visualization tool that allows users to display time series such as EEG, EMG, and ECG, using a set of displaying parameters (Table 2).

**Table 2.** Display parameters for capturing each image.

| Parameters | Description |
|---|---|
| Timeline | 10 s/page |
| Amplitude | 100 µV |
| Background | White |
| Axes | Hidden |
| Channels | 19 |
| Channel nomenclature | Visible |
| Electrooculogram | Visible |
| Electrocardiogram | Visible |
| Tonality | Blue |
| Edges and software interface | Hidden |

Four hundred one images with the signals were extracted, and they were organized into normal and abnormal, each one, in jpg format with $1920 \times 906 \times 3$ pixels. Figure 3 presents an example of the extracted images. One-hundred EEGs were taken, and each EEG had an approximate duration of 30 min, for which each EEG had approximately 180 images because each image has a period of 10 s of the examination. Not all 180 images were considered from each EEG, only the images with epileptiform events identified by the neurologist, and some randomly selected normal images. For patients with normal EEG recordings, all the images considered were randomly chosen.

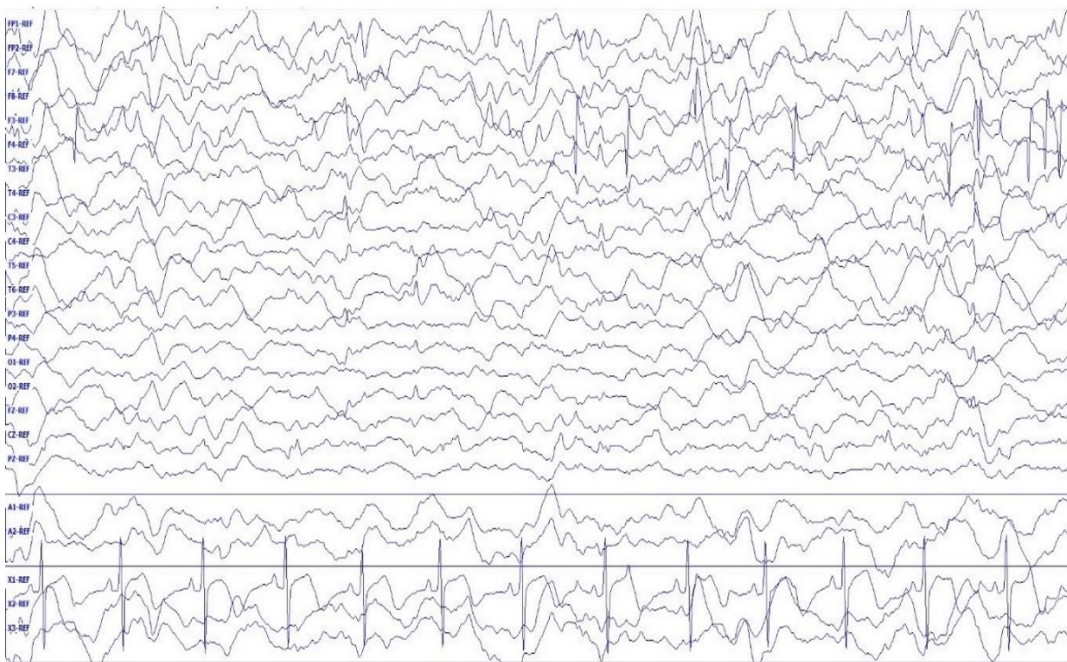

**Figure 3.** Example of an EEG page annotated as abnormal.

The distribution of normal and abnormal EEG is presented in Table 3. Concerning the abnormal category, 166 images were gathered in total.

**Table 3.** Dataset composition according to diagnosis.

| EEG | No. Patients | EEG Pages |
| --- | --- | --- |
| Normal | 66 | 235 |
| Abnormal | 34 | 166 |

### 2.1.3. Model Selection

A classification system of transfer learning was developed by two convolutional neural networks (CNN), AlexNet [32] and GoogLeNet [33]. Their architectures contain 8 and 22 deep layers, respectively; these were initially trained to classify up to 1000 different image classes [33,34], with millions of images from the ImageNet database (ImageNet). Although other CNN such as ResNET, Inception-ResNET, and NASNetLarge were evaluated, these did not achieve good results. Therefore, they were not considered in this report.

### 2.1.4. Model Fine-Tuning

The model fine-tuning, also known as data tuning, was performed by following the partial fine-tuning method, which consists of freezing the first layers of the model and retrains with the specific dataset layers. In this case, it was replaced with a new fully-connected layer with 31 outputs and a new one with two class labels corresponding to normal and abnormal (seizure) pages, respectively; this is binary classification. In such a sense, and considering the requirements of the CNN, images were resized to $224 \times 224 \times 3$ pixels; those resized images served as input to both pre-trained networks and, to discover seizures, the last fully connected layer and the final classification layer of the networks were replaced.

### 2.1.5. Model Evaluation

In this step, the k-fold method was used with cross-validation to obtain the best model from different performance indices. This methodology guarantees that the results are independent of training and testing datasets [18]. First, a dataset partition was performed.

EEG images of 70% of the patients were randomly chosen to train, and 30% of the remaining patients were used as a held dataset to assess the model's performance; 401 images were considered in the model evaluation. Finally, the confusion matrix was calculated, and, from there, the following indices were derived: Accuracy, Sensitivity, Specificity, Precision, F1-Score, and Matthews Correlation Coefficient (CCM).

### 2.2. Second Stage: Multi-Class Classification

In this second stage, similar steps presented in the first stage were performed. However, there were some changes, as described below.

In data preparation, the abnormal signals showed six patterns: sharp waves, spikes, poly-spikes, spike-and-wave, periodic, and above combinations. These, plus the normal class, are the seven final classes considered at this stage. Figure 4 presents three types of abnormal signals: (a) spikes, (b) poly spikes, and (c) spike-and-wave. Additionally, this figure illustrates a normal signal (d). The image dataset used in this work has a considerable number of spikes and poly spikes, which they will be the types of abnormalities to consider in this work.

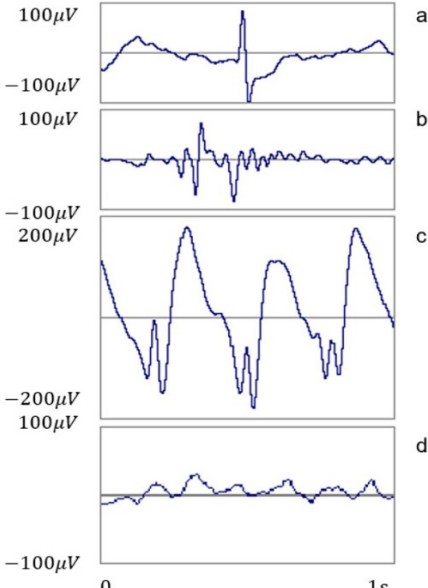

**Figure 4.** Examples of abnormal and normal signals: (**a**) spikes, (**b**) poly-pikes, (**c**) spike-and-wave, and (**d**) normal.

The number of selected images was increased in the model evaluation, from 401 images used in stage one to 601 images. Out of these 601 images, 435 were normal and 166 abnormal out of the 166 abnormalities, 35 poly-spikes, 50 spikes, 35 spike-and-wave, 11 sharp waves, 5 periodic, and 30 combinations.

Additionally, in this second stage, the six types of abnormalities (and the normal class) were initially considered in model evaluation. Later, the number of classes was decreased in each evaluation iteration with each of the CNN to obtain models with 6, 5, 4, and 3 abnormalities. This process allowed us to evaluate if the performance indices (accuracy, precision, etc.) were affected when considering a more significant number of classes and the balance of the dataset. Finally, the complete results in the training and evaluation stage (with all the performance indices) were calculated for the model with the number of classes with better results in accuracy (four classes: three abnormalities and normal) and the worse outcomes inaccuracy (seven classes: six abnormalities and normal).

### 3. Results

The results obtained in each stage are presented in this section.

### 3.1. Results in Binary Classification

3.1.1. Modeling: Retraining of the Convolutional Neural Networks (CNN)

In this step, AlexNet [31] and GoogLeNet [32] CNN were partially varied and trained repeatedly, as indicated in Section 2.1.4. Furthermore, the cross-validation with the k = 10 method was assumed to ensure independent results of the data partition and higher performance. Results for the training stage, for each CNN in terms of accuracy, are presented below.

AlexNet. As a first step, it was necessary to retrain the Alexnet by using 281 images of EEG pages (70% of 401 images), a batch size of 10 and 15 epochs. The final output made a 95% accuracy possible, as measured during each iteration until reaching the 15 epochs. Figure 5 presents the training results with AlexNet.

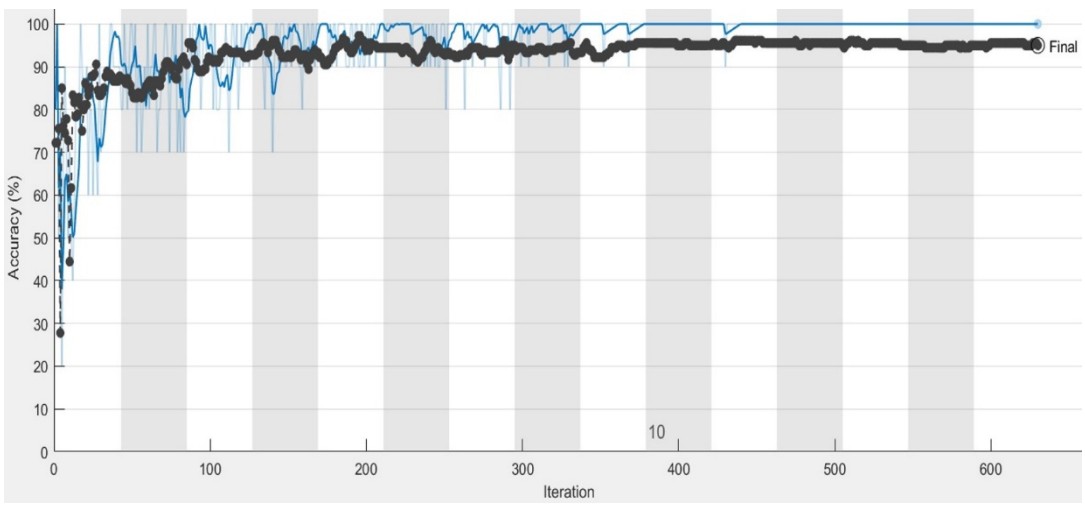

**Figure 5.** Accuracy AlexNet during training, trained using 15 epochs.

GoogLeNet. As a second step, it was necessary to retrain the GoogLeNet by using the same 281 images of EEG pages (70% of 401 images), a batch size of 10 and 15 epochs. The final output made accuracy higher than 93%, as measured during each iteration until reaching the 15 epochs. Figure 6 presents the training results with GoogLeNet.

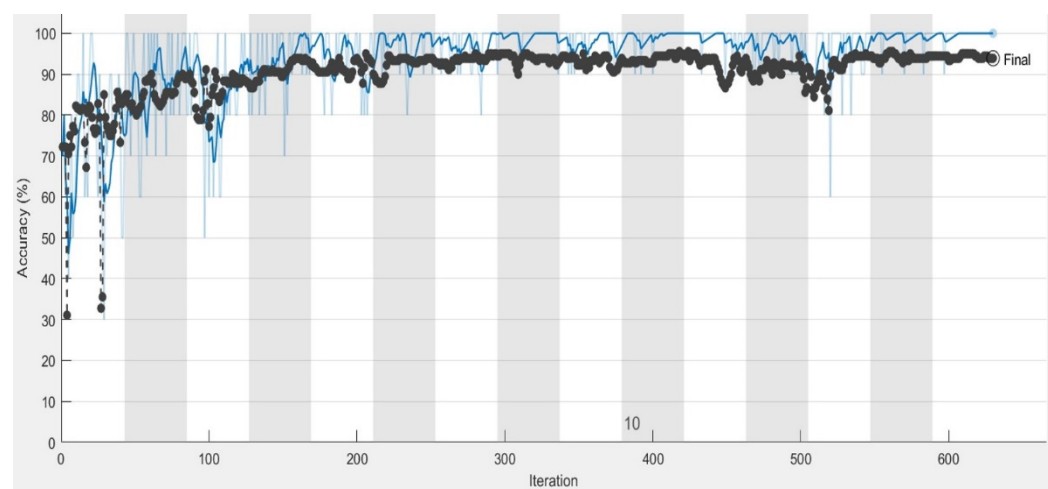

**Figure 6.** Accuracy of GoogLeNet during training, trained using 15 epochs.

3.1.2. Evaluation

The test data set consisting of 120 held images (30% of 401 images), 33 EEG pages with seizures, and 87 EEG pages without seizures were used to estimate the performance of this

model. Regarding CNN (AlexNet and GoogLeNet), a confusion matrix was employed for the results shown in Table 4.

**Table 4.** Confusion matrix for test data in binary classification.

| | Prediction | AlexNet | | GoogLeNet | |
|---|---|---|---|---|---|
| | | **Abnormal** | **Normal** | **Abnormal** | **Normal** |
| True Class | Abnormal | 28 | 5 | 27 | 6 |
| | Normal | 1 | 86 | 1 | 86 |

Although the data set was unbalanced, the performance of both classifiers was optimal. Furthermore, some seizures were identified through many mistakes caused by the unbalanced dataset.

It was possible to calculate the performance indices by using the confusion matrix values. The values are presented in Table 5.

**Table 5.** Performance indices are obtained from the confusion matrix in binary classification.

| Index | AlexNet (%) | GoogLeNet (%) |
|---|---|---|
| Sensitivity/recall | 96.55 | 96.43 |
| Specificity | 94.51 | 93.48 |
| Precision | 84.85 | 81.82 |
| Accuracy | 95.00 | 94.17 |
| F1 Score | 90.32 | 88.52 |
| MCC | 87.30 | 85.16 |

The trained model was able to generalize due to the implemented data set because it presented outstanding performances compared to new data not included in the training stage.

The model presented a performance of approximately 95% (and the rest of the performance indexes above 87.30%). These results demonstrated that identifying epileptiform events from raw EEG images combined with deep learning techniques is feasible.

Another important aspect of the results of this classification is the execution time of the algorithm. Such time is only a few minutes (less than 5 min); therefore, with this technique, the initially stated objective of improving the efficiency of the epilepsy diagnostic process is achieved, reducing the time necessary to review the EEG exam by part of the neurologist, and reducing the number of pages he needs to review.

### 3.2. Results in Multi-Class Classification

As previously mentioned, the classes considered in this classification were initially six abnormalities (sharp waves, spikes, poly-spikes, spike-and-wave, periodic, and combinations of the abnormalities) and the normal class (seven classes in total). Subsequently, 6, 5, 4, and 3 classes were evaluated, considering only the performance index "accuracy" (maintaining in each option the normal class). Finally, as mentioned in Section 2.2, the number of images considered was increased to 601 images. Final classification was performed using both CNNs. However, AlexNet outperformed over GoogLeNet. Figure 7 shows the results obtained of the accuracy in each of the classifications made for AlexNet.

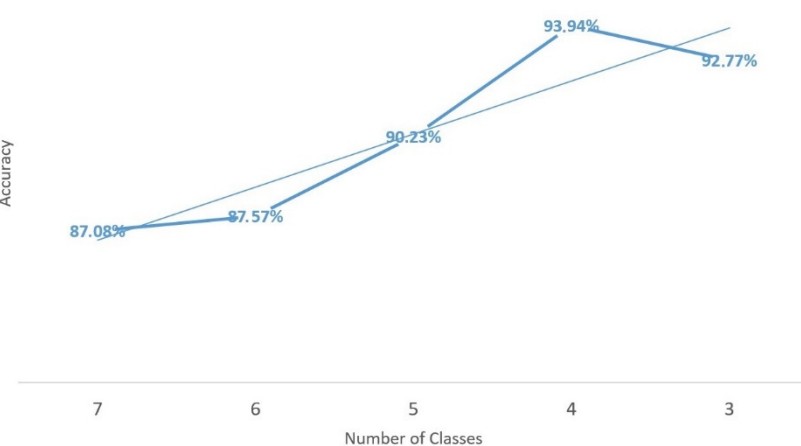

**Figure 7.** Accuracy of multi-class classifications (seven to three classes).

As expected, as the number of classes considered was decreased, performance increased (in most cases because the result with three classes was lower than the result with four classes). Therefore, the straight line that was graphed in Figure 7 corresponds to the increasing trend.

The training performed using four classes (spikes, poly-spikes, spike-and-wave, and normal), and 421 images (70% of 601 images) is presented in Figures 8 and 9, where accuracy and loss (respectively) are calculated.

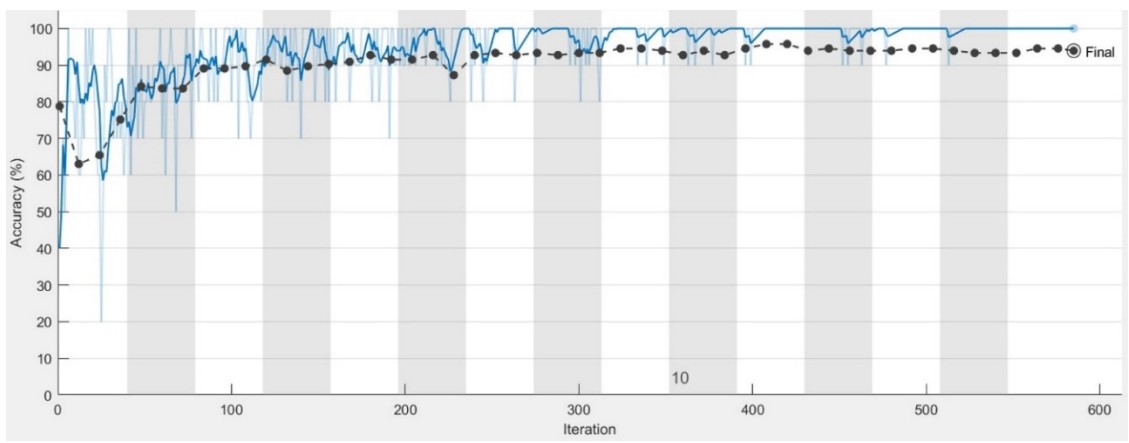

**Figure 8.** Accuracy during training (using four classes for classification).

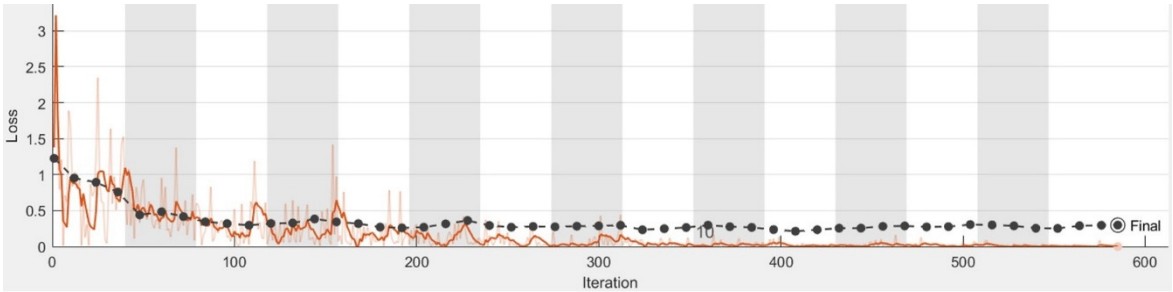

**Figure 9.** Loss during training (using four classes for classification).

In evaluating four classes of classification, a test data set of 180 held images (30% of 601 images). Regarding the CNN (AlexNet), a confusion matrix was employed for the results, as shown in Table 6.

**Table 6.** Confusion matrix for four classes classification.

| | | Prediction | | | |
|---|---|---|---|---|---|
| | | **Normal** | **Poly-Spikes** | **Spikes** | **Spike-and-Wave** |
| True Class | Normal | 129 | 0 | 1 | 0 |
| | Poly-spikes | 0 | 7 | 2 | 1 |
| | Spikes | 2 | 0 | 8 | 0 |
| | Spike-and-wave | 2 | 2 | 0 | 11 |

The "accuracy" index results were reasonable considering the results obtained in other works (e.g., [21]). However, we consider it pertinent to calculate the other performance indices for the four and seven classes under evaluation, where the best and worst accuracy results were presented.

The values obtained for all performance indices in this evaluation (using 4 and 7 classes) are presented in Table 7.

**Table 7.** Performance indices obtained from the confusion matrix in multi-class classification.

| Index | Seven Classes | Four Classes |
|---|---|---|
| Sensitivity/recall | 65.81 | 81.75 |
| Specificity | 68.97 | 80.61 |
| Precision | 67.11 | 80.25 |
| Accuracy | 87.08 | 93.93 |
| F1 Score | 67.51 | 80.75 |
| MCC | 69.30 | 82.70 |

Finally, it was considered relevant to analyze the normal signals for patients diagnosed with epilepsy and those ruled out for this disease. Therefore, the normal signals without epileptiform events were divided into patients without epilepsy and patients with epilepsy. The two networks (AlexNet and GoogleNet) were retrained with 95.56% and 91.12% (respectively). AlexNet training is presented in Figures 10 and 11.

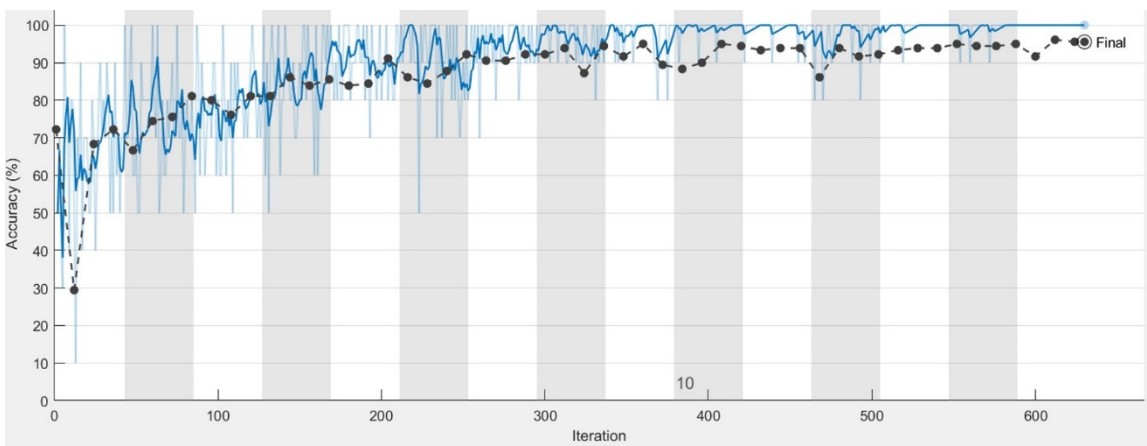

**Figure 10.** Accuracy during training (classification of normal signals).

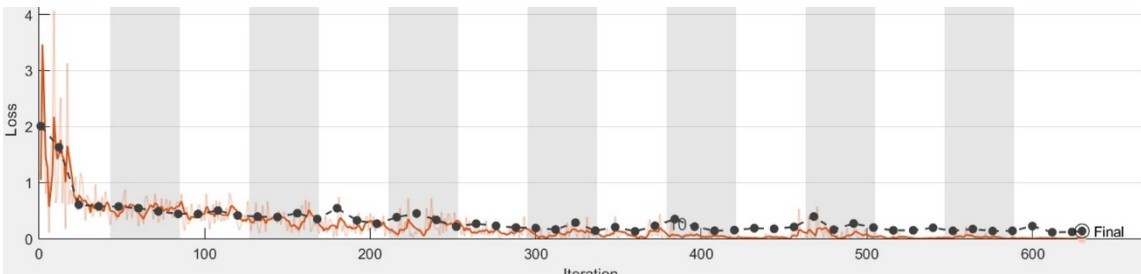

**Figure 11.** Loss during training (classification of normal signals).

The confusion matrix obtained in evaluating this last classification of normal signals is presented in Table 8. The accuracy obtained in the evaluation of this classification was 93.08.

**Table 8.** Confusion matrix for test data in classification of normal signals.

|  |  | Prediction | |
|---|---|---|---|
|  |  | **Patient with Epilepsy** | **Patient without Epilepsy** |
| True Class | Patient with epilepsy | 45 | 5 |
|  | Patient without epilepsy | 5 | 125 |

We considered the result obtained in this last binary classification of normal signals necessary to detect the possible presence of the disease (epilepsy) in signals without any apparent type of epileptiform event (which could be easily confirmed by a neurologist). This result could be relevant considering that no epileptiform event may occur in an EEG (30 min approx.) even though the patient has the disease. We consulted about literature that had performed this type of classification, but it was not possible to find results in this regard.

A heat map was created to analyze if the results obtained in the last classification of normal signals (patients with epilepsy and patients without epilepsy) were relatively logical. This map made it possible to determine the image areas that the algorithm used to determine the respective classification. In Figure 12, some examples of the obtained heat map images are presented. The areas in red, or close to this color, are those that the classifier indicated that is an image of a possible patient with epilepsy. The images present the signals obtained in the different channels, in black lines, and the respective colors accepted. Unfortunately, the resolution of these images cannot be improved because the use of the transfer learning technique requires a resize of the images to analyze them

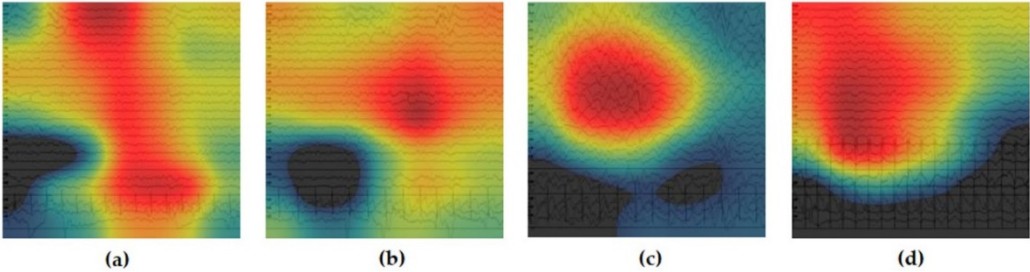

**Figure 12.** Heat maps in EEG images for binary classification of normal signals, (**a**–**d**) are examples of the various types of heat maps that were obtained.

## 4. Discussion

The proposed system allows the incorporation of a technological tool for the analysis of EEG exams, intending to support the epilepsy diagnosis process performed by neurologists. Through this tool, it is possible to identify, in an EEG exam, the segments of the signals that

are possibly related to an "epileptic seizure". Therefore, a considerable percentage of the EEG exam is discarded, which the neurologist will not have to review since it contains no signals that are of interest. By reducing the number of pages (or images) of the EEG exam that the neurologist must review, the time to make a diagnosis of the disease decreases considerably. This reduction in diagnosis time improves the sustainability of the service (in social and economic aspects), allowing an increase in the number of patients that could be diagnosed and reducing associated costs, considering the reduction in the time required by the specialist.

This methodology in support of the specialist could be useful for more accurate work on the monitoring of crises in patients who no longer take drugs (antiseizure medications). It was seen that the most important risk factors for withdrawal failure are the etiology of the epilepsy syndrome and epilepsy-related factors, worsening or persistence of epileptiform abnormalities on EEG recordings at the time of discontinuation or during drug tapering, and brain Magnetic Resonance Imaging (MRI) abnormalities [35].

The results obtained in our research with respect to the detection of events of interest in the EEG exam are reasonable when compared with those obtained in similar works [21,24]. The work done by Gómez et al. only detects "epileptic seizures" in general [21]. It performs a binary classification (normal or abnormal) in an EEG image. It is not multi-class like the one by Raghu et al. [24], and our research, which performed both types of classification (binary and multi-class). Gómez et al. work has excellent accuracy and specificity of 98–99% (in both indices, in both datasets, evaluated). However, it does not have good results concerning the other performance indices such as precision, recall, and F-measure (62.7%, 58.3%, and 59.0%, respectively, in the best-evaluated model). The results in our work are lower in accuracy and specificity (93–95%) in the binary classification (normal or abnormal signals in each image). Still, it has much better results in the other indices: precision is 84.85%, recall 96.55%, and F-measure 90.32% in the better model (AlexNet CNN). Therefore, it is considered that it is better to have all the indices above 84.85% than to have only some between 98% and 99% and some others between 58.3% and 62.7%.

The work of Raghu et al. proposes a multi-class classifier, detecting seven types of abnormalities in an EEG image, with a maximum accuracy of 82.85% [24]. The accuracy in the multi-class classification presented in our research was 87.08% using seven classes (six abnormalities and normal class) and 93.93% using four classes (three abnormalities and normal class). Although the accuracy index is better in our research, it should be noted that, in Raghu's work, an additional abnormal type is used, and the results obtained in the other performance indices are not presented. In contrast, our research offers all the indexes for the multi-class classification of 4, and seven classes. In the 4-class classification, the accuracy index was between 80.25% and 93.93%. While in the classification of seven classes, the index was between 65.81% and 87.08%.

In Raghu's work, 10 CNNs were used, while in our research, only five were used, of which the results of two of them (AlexNet and GoogleNet) are presented because the results obtained with the other three were not good enough.

Another highlight in our research is the data set collected since the project related to this work performs the collection and processing work. The number of patients and the number of EEGs is remarkable concerning other related works (not specifically concerning [21,24], but about other works mentioned in such references and different results in literature). Additionally, pediatric EEG (used in this research) presents more significant variability than adult EEGs. Therefore, they have a greater level of difficulty in interpreting their signals.

The results obtained in the binary classification of normal signals had good results (93.08% accuracy). However, it was not possible to compare it with similar works because they were not found. Usually, the identification of an EEG of a patient with epilepsy is performed through epileptic seizures (abnormal signals). The approach of identifying it through normal signals was not found in previous works. The revision of heat maps by a neurologist is the next step to identify patterns.

Finally, regarding the limitations of this work, these are related to the number of types of epileptiform events detected, the number of EEG exams performed, and the patients who participated in the study. Although the six types of epileptiform events that occur most frequently were considered, it is important to identify a larger number, trying to cover all those that could occur, to improve support for the diagnosis of the disease. The number of EEG exams and patients is considerably high and increasing them would help to improve the training process of the algorithms used, and possibly improve the results obtained.

## 5. Conclusions

These results demonstrated that identifying epileptiform events from raw EEG images combined with deep learning techniques (such as transfer learning) is feasible. This identification of epileptiform events allows an improvement in the epilepsy diagnosis process, supporting the work performed by neurologists, by reducing the number of images to be reviewed in the EEG.

This work proposes a machine learning pipeline capable of detecting and identifying pages of EEG examinations with different abnormalities useful for an epilepsy diagnosis. These abnormalities include sharp waves, spikes, poly-spikes, spike-and-wave, periodic, and combinations of these abnormalities. In addition, an approach based on digital image processing and computer vision was introduced, which is a novel approach. It is an alternative to classic signal processing and feature engineering proposed in other approaches to EEG signal analysis.

The abnormalities detection in this work was performed using two approaches in two different stages. In the first stage, a binary classification approach was used (normal or abnormal), in which excellent results were obtained, with all the performance indices above 84%. These results are positive, since, although other studies reported better indicators of accuracy and specificity, the values of the different indexes (precision, recall, and F-measure) were lower than those obtained in this study. In the second stage of this work, a multi-class classification was performed, where up to six classes of abnormalities were considered in the EEGs. The detailed results of this type of classification were presented for four classes (3 abnormalities and normal class) and seven classes (6 abnormalities and normal class). These results allowed for showing better results of index performance compared to previous works reviewed in the literature.

As an additional result to the two phases of the work mentioned, a binary classification was performed to determine within the normal signals which belongs to a signal from a patient with epilepsy or a patient without epilepsy. The results obtained have a good level of accuracy, although it was not possible to find similar works to compare the results. Furthermore, this approach was extended to detect specific locations within the image where seizure is presented, using heat maps as support. We think it is an exciting approach to support the diagnostic process of epilepsy in EEG, which should be evaluated in more detail in future works.

EEG abnormal events were identified in this work using the two aforementioned approaches. However, different types of epilepsy were not identified. The most well-known types of epilepsy are: focal, generalized, focal and generalized, and unknown. The neurologist who reviewed the EEG exams, in this work, identified the zones in EEG where an epileptiform event was present but was not asked to identify the type of epilepsy. The identification of types of epilepsy in EEG is proposed as future work, which can help to better diagnose the disease.

The methodology proposed in this work needs to be tested in other databases with which other comparisons can be made. Finally, after complying with the previous steps, this methodology can be adapted to be implemented in a clinical setting for the semi-automatic detection of epilepsy. Furthermore, the proposed method is expected to impact the diagnosis and treatment of epilepsy patients significantly, for example, in low-income countries with high patient volumes and regions with limitations in the provision of neurology services.

**Author Contributions:** Conceptualization, D.M.L., R.S.-C. and R.V.-C.; methodology, D.M.L., R.S.-C. and R.V.-C.; software, M.S.M., C.E.S.T. and R.V.-C.; validation, M.S.M., C.E.S.T. and R.V.-C.; formal analysis, D.M.L., R.S.-C. and R.V.-C.; investigation, M.S.M., C.E.S.T. and R.V.-C.; resources, D.M.L., R.S.-C. and R.V.-C.; writing—original draft preparation, M.S.M., C.E.S.T. and R.S.-C.; writing—review and editing, D.M.L., R.S.-C. and R.V.-C.; supervision, D.M.L. and R.V.-C.; funding acquisition, D.M.L. and R.V.-C. All authors have read and agreed to the published version of the manuscript.

**Funding:** This research received no external funding.

**Institutional Review Board Statement:** The study was conducted according to the guidelines of the Declaration of Helsinki, and approved by the Ethics Committee for Scientific Research of Universidad del Cauca, Colombia (as stated in minutes No. 10 of 24 June 2015) due to it adjusts to the bioethical requirements of the Colombian Ministry of Health as stipulated in Resolution 008430 of 1983.

**Informed Consent Statement:** Informed consent was obtained from all subjects involved in the study.

**Data Availability Statement:** The data presented in this study are available on request from the corresponding author.

**Acknowledgments:** The authors wish to thank Universidad del Cauca (Telematics Department and Physics Department). The Colombian Agency funded NeuroMoTIC project for Science, Technology, and Innovation—MINCIENCIAS—in call 715–2015, "Call for Research and Development Projects in Engineering" Project "NeuroMoTIC: Mobile System for Diagnostic Support of Epilepsy", contract No. FP44842–154–2016.

**Conflicts of Interest:** The authors declare no conflict of interest.

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
