# Peer review of "Digital Transformation in Epilepsy Diagnosis Using Raw Images and Transfer Learning in Electroencephalograms"

_sustainability, doi:10.3390/su141811420_

Round 1
Reviewer 1 Report
The work is very interesting, I have some suggestions for the authors: it should be specified how the EEG data were collected: in sleep or wakefulness. It would be interesting to know if a classification of type of epilepsy has been made in patients with EEG abnormalities. This methodology in support of the specialist could be useful for a more accurate work on the monitoring of crises in patients who no longer take the drug, in fact in a review it was seen that the most important risk factors for withdrawal failure are the etiology of the epilepsy syndrome and epilepsy-related factors, worsening or persistence of epileptiform abnormalities on EEG recordings at the time of discontinuation or during drug tapering, and brain MRI abnormalities.
Other suggestions concern the layout: the sentence "This stage's methodology consisted of five steps (Figure 2): data collection, data preparation, model selection, model fine-tuning, and model evaluation" has been written twice (lines 166-167 and 171-172). Change the formatting from line 304-314. Change the references in the text according to the guidelines of the newspaper “In the text, reference numbers should be placed in square brackets [], and placed before the punctuation; for example [1], [1–3] or [1,3] "
Reviewer 2 Report
I have read the MS “Digital Transformation in Epilepsy Diagnosis Using Raw Images and Transfer Learning in Electroencephalograms” and found it to be very interesting. Most of the MS parts are well-balanced and well-written. I recommend publishing the MS with minor revisions.
1- What are the limitations of this study? Please give a mention of these at the end of the discussion section.
2- I recommend reading and citing the following paper as will: Türk, Ö.; Özerdem, M.S. Epilepsy Detection by Using Scalogram Based Convolutional Neural Network from EEG Signals. Brain Sci. 2019, 9, 115. https://doi.org/10.3390/brainsci9050115
